# Evidence from a Choice Experiment in Consumer Preference towards Infant Milk Formula (IMF) in the Context of Dairy Revitalization and COVID-19 Pandemic

**DOI:** 10.3390/foods11172689

**Published:** 2022-09-03

**Authors:** Jing Zhang, Scott Waldron, Xiaoxia Dong

**Affiliations:** 1School of Agriculture and Food Sciences, The University of Queensland, Brisbane 4072, Australia; 2Agricultural Information Institute, Chinese Academy of Agricultural Sciences, Beijing 100081, China

**Keywords:** China, infant milk formula, consumer preference, choice experiment

## Abstract

China is the largest global consumer of infant milk formula (IMF). Chinese consumer preferences towards IMF have evolved over time but have also been rocked in recent years by COVID-19 with major implications for the IMF industry, globally and within China. This study is the first to document parents’ preferences toward IMF since the outbreak. We used novel methods to do so, through an online choice experiment of 804 participants that included risk perceptions and socio-demographic variables. Our study finds that Chinese parents continue to prioritize quality and safety attributes of IMF represented by functional ingredients, organic labelling and traceability information. Notably, it also finds greatly increased confidence in Chinese domestically produced IMF and an underlying preference away from expensive products. This implies that the era of ‘go for foreign’ and ‘go for the most expensive’ in IMF purchasing may be coming to an end. The shift in sentiment is driven by the longer-term revitalization of the Chinese dairy industry, accelerated by COVID-19. Understanding these trends will be of major benefit to both Chinese producers and non-Chinese exporters of IMF.

## 1. Introduction

In an attempt to alleviate concerns about an ageing population and falling birth rate, China ended its one-child policy in 2015 to allow all couples to have two children and further relaxed the limit to three children on 31 May 2021 [1]. This was followed by incentive measures including tax deductions, childcare services, financial subsidies, parental leave and the increased protection of women’s rights in employment [2]. Given the very low exclusive breastfeeding rate in China [3], more babies in China could be expected to mean more babies drinking infant milk formula (IMF), which is one of the most in-demand and frequently purchased childcare-related products in the country for a range of social, economical and cultural reasons [4,5].

China relies heavily on imported IMF, especially after the melamine contamination scandal that engulfed the domestic industry in 2008. Consumers have subsequently prioritised product quality and safety as the most important factors when shopping for baby food [6]. Imported IMF, serving as a proxy for quality and safety, was highly valued in Chinese consumer preferences and decisions in the pre-COVID-19 era [7,8]. Particularly with the introduction of the universal two-child policy in 2015, China’s IMF imports almost doubled from 179,897 tons to 356,382 tons in 2019, taking around 50% of global export, favoring mostly European brands as well as products from New Zealand and Australia [9]. However, growth ground to a halt after 2019 as China’s global imports of IMF declined by 3% in 2020 and a further 22% to 272,669 tons in 2021. Imports from the two largest global sources—Netherlands and New Zealand—dropped 19 and 16% respectively, while other countries dropped 20–40% in volume over between 2020 and 2021 [9]. Studies have not revealed whether it is a temporary disruption due to COVID-19 or a shift in consumer preferences.

Information on consumer preferences is central to understanding and predicting China’s import demand for IMF and its associated ingredients such as whey powder. However, there are few empirical studies on parent preferences for IMF in the latest literature, particularly in post COVID-19 pandemic era. The pandemic has also revived debate globally and in China about the re-shoring of industries [10,11]. Consumer preferences are a key aspect of this debate, with major implications for IMF and its associated products both in and outside of China. 

The study had a two-step objective: to explore the attributes that influence Chinese parents’ purchase decisions of IMF; and to understand possible non-product reasons for the preferences, especially given the uncertainty brought about by the COVID-19 pandemic. The rest of this paper is organized as follows. Section 2 provides a review of literature followed by an explanation of choice experiment methodology including experimental design, data collection and modelling. Research findings and discussion are presented in Section 4 and Section 5, followed by conclusions.

## 2. Literature Review

Understanding consumer preferences is a fundamental aspect of businesses decision-making including developing and delivering products to meet demand, as well as marketing and pricing. A range of stated preference approaches have been used to understand consumer preferences including conjoint analysis [12,13], contingent valuation [14,15], choice experiments [16,17] and (laboratory) experimental auctions [18,19,20]. Contingent valuation is widely used in nonmarket valuation especially in the areas of environmental and ecological impact assessment [21]. In the field of market valuation, experimental auctions are commonly used by setting up a real environment such as supermarket to enable participants to consecutively bid on real products rather than hypothetical products [19]. A conjoint analysis approach typically requires participants to rate each product attribute individually, while choice experiments (also known as choice based conjoint analysis) present multiple sets of product attributes and ask respondents to choose the set they prefer. This method is generally considered most consistent with an actual shopping experience [22] and was therefore selected for this study. 

All techniques are regarded as experimental and subject to criticism on their hypothetical biases and the reliability of results [23]. To avoid the hypothetical biases and improve the external validity, Haghani et al. [24,25] suggest a set of ex-ante strategies for choice experiment studies in the field of consumer economics, namely ensuring incentive compatibility, providing an opt-out option and making the choice setting as tangible as possible. These measures were adopted in this study. For instance, survey participants were paid, but not if there were indications of a low-quality response, indicated by the completion of the survey in less than five minutes or of repetitive patterns. Opt-out options were provided (“I would not buy either of them”) and choice alternatives were presented in pictures of IMF tins rather than more abstract words in table. In addition, the reliability of results can also be mitigated by cross-verification of two or more modelling techniques such as an ex-post strategy [26,27,28]. This study does so by applying two independent logit (conditional logit and random parameter logit) models. 

A number of themes emerge in the more specific literature on consumer preferences towards IMF products. One is that a successful choice experiment requires the identification of the most important and relevant attributes. The presence of either too many or too few attributes may lead to unreliable results [29]. The main IMF attributes used in previous studies include the country of origin, price, nutritional information (probiotics contained), brand recognition, packaging, quality certification, organic certification and traceability systems [7,8,30,31,32,33,34]. Choice experiments in this study selected the relevant attributes of nutritional information, organic certification and traceability codes as indicators of product quality and safety, which has been confirmed by most studies above. Two other primary attributes—price and origin—were of particular interest in the context of COVID-19. Multiple studies conducted before 2020 show that Chinese consumers lack confidence in the safety of domestically produced IMF, and mistrust domestic IMF supply chains, which means that consumers often use the country of origin and price to infer the quality of a product [7,8]. However anecdotal evidence and various studies [35,36,37,38] suggest that consumer preferences, perceptions and habits may have changed in the era of COVID-19. This paper aimed to provide up-to-date empirical evidence on current consumer perceptions and confidence to IMF in the COVID-19 era.

## 3. Materials and Methods

### 3.1. Choice Experiment Design

In the context of this study, IMF is viewed as a collection of product attributes from which consumers derive utility. Choice experiments enable the target consumers to evaluate trade-offs among attributes by replicating real-life purchasing situations and allowing the evaluation of multiple attributes. As one of the key issues in designing a choice experiment, attributes and their corresponding levels set in this study were extracted from prior studies (as shown in Table 1), but were tailored in this study to preferences toward IMF in particular. Two four-level and three two-level IMF attributes were included including product origin, price, organic label, functional ingredients and traceability information. Detailed information regarding the specific attributes and their corresponding levels is presented in Table 1. The full list of attributes was tested in a pilot survey. Some were dropped because they overlapped with others or were identified as being relatively unimportant (e.g., packaging and taste).

The combination of attributes and levels in Table 1 form a total of 4^2^ × 2^3^ = 128 virtual product profiles, which is too many to request respondents to compare and select from. A total of 15 to 20 profiles will fatigue the consumer [47]. Therefore, to accommodate our study objective while maximising the statistical performance of coefficient estimates, we employed a fractional factorial orthogonal design to ensure attribute level balance over alternatives using SPSS 28 software. This generated 16 alternatives for a single block of eight choice sets. Each participant was presented with the same eight choice sets (without multiple versions) and was asked to choose one and only one option out of two IMF packages and an opt-out option “I would not buy either of them” to better simulate an IMF purchasing decision in each choice set. Pictures of tins of IMF featuring the attribute levels were used to represent the alternatives (see Figure 1 for a sample choice set). Prior to being presented with the choice sets, respondents were fully informed of the definition of each of the quality attributes and their corresponding levels. The respondents were also informed that, except for these attributes in a choice set, the IMF products presented had no difference in appearance, taste, grade, or any other attributes. To control for hypothetical bias, a note was added for respondents to assume that the product is for their own child, that they apply real budgetary constraints and that they have the option of not purchasing any IMF in a choice set. Additional information on participants’ socio-economic demographics, consumption habits, risk perceptions and attitudes toward products currently in use were also collected after the choice experiment.

### 3.2. Survey Design and Data

Face-to-face interviews are currently difficult to conduct in China because of the health concerns of COVID-19, social distancing requirements and the “zero COVID” and “dynamic clearing” policies (The “zero COVID” and “dynamic clearing” policies refer to stopping transmission of the coronavirus through lockdowns, mass testing and quarantines to achieve zero cases). The survey was therefore necessarily administrated online, which has several additional benefits. Online surveys for choice experiments are likely to elicit a more thoughtful and real-life response than face-to-face surveys due to social desirability bias and interviewer effects [48]. A pilot survey was conducted through personal contacts (“Friends Circle”) on Wechat App (a popular social media platform in China) and feedback was incorporated to ensure the clarity of attributes/levels and understanding of each statement to the respondents. The formal questionnaire was distributed to the targeted population with child(ren) younger than three years old through an online platform called Agri-watch (http://www.agriwatch.cn/, accessed on 21 August 2022). Participants were paid a small incentive fee (CNY20/person, equivalent to around USD3/person) to complete the questionnaire. In an attempt to obtain reliable data, the questionnaire started with a clear description of the objectives and instructions of the study and concluded with a specific request for each participant to take a real-time photo of the IMF package currently consumed by their baby at home. The questionnaire was available online for two weeks in April 2022, resulting in a final sample of 804 responses after removing the responses that took less than five minutes to answer. 

### 3.3. Econometric Modelling

The choice experiment methodology is underpinned by two core consumer theories—Lancaster’s characteristics demand theory [49] and McFadden’s random utility theory [50]. The former states that consumers usually derive utility from the characteristics of a product rather than the product itself. This study presented the IMF product to respondents as different combinations of five key component attributes (Origin, Functional, Organic, Traceable and Price). The latter theory states that consumers generally choose the product that carries greater utility based on the values derived from different combinations of product attributes. Given the existence of unobserved and random factors in the decision-making process, it is difficult to predict with certainty that the consumer would select a particular alternative, while it is possible to estimate the probability of that the perceived utility of one alternative is greater than that of all the other available alternatives. Lancaster’s characteristics demand theory and McFadden’s random utility theory allowed this study to divide the respondent’s utility into an observable deterministic component and an unobserved random component as shown in Equation (1) below.
U_in_ = V_in_ + ε_in_ = α′ X_in_ + β′ Z_in_ + ε_in_(1)
where U_in_ denotes the utility of IMF alternative i in a choice set responded by individual n; V_in_ denotes the observable deterministic component that specified as a function of alternative-specific attributes vector X_in_ and case-specific demographic and perception vector Z_in_, estimating how consumers’ preferences vary with different levels of demographic and perception variables; α′and β′ are the vector of coefficients describing the marginal utility of the attributes vector as well as demographic and perception vectors; ε_in_ is a random variable that accounts for the effects on preferences of unobserved attributes of the alternative i and individual n. Then, the probability that the individual n chooses alternative i over any other options j from a given choice set is: P_in_ = P (U_in_) > P (U_jn_) for all j ≠ i(2)

The estimation of choice probabilities P_in_ differs with the distribution of unobserved random components. Conditional logit (CL) and random parameter logit (RPL) were applied in this study as two alternative approaches to account for differences in consumer preferences, where CL assumed the random components are independently and identically distributed (IID) with the implication that alternatives have independence from irrelevant attributes, while RPL, also known as the mixed logit model, is undertaken in case the IID assumption is violated. Conditional logit (CL) and random parameter logit (RPL) models were conducted through the commands of “clogit” and “mixlogit” in Stata 14.2.

## 4. Results

### 4.1. Consumer Characteristics

Descriptive statistics of selected demographic characteristics for the final sample are presented in Table 2. As could be expected, a greater number of responses were obtained from female than male respondents, as mothers are most likely to be more interested in and to make the decisions on IMF selection. The age of the respondents was concentrated in the 20 to 40 years old range, with a family size of three to four persons. Compared to the Chinese population, the sample had a higher percentage of individuals holding a bachelor’s degree or above (65.2%). The sample also had a higher percentage of families with above national average family income; 89.3% of the respondents’ reported a yearly family income of more than CNY 100,000, which is close to urban statistics of family of three in 2020 [51].

Consumers’ perception on product quality and safety were asked in the form of how much you disagree or agree with each of these statements by using seven-point Likert scale of one to seven where one means strongly disagree and seven means strongly agree. As shown in Table 2, 40.6% agreed (scale > 4) that the quality and safety of domestically produced IMF is reliable, 31.7% disagreed (scale < 4) that the quality and safety of domestic produced IMF is reliable, while 27.7% neither agreed nor disagreed (scale = 4). Respondents were also asked questions on the performance and strictness of the system that oversees quality and safety standards form domestic IMF (i.e., regulation, testing, inspection and enforcement). A proportion of 60.6% of respondents perceived the system to be strict, 23.8% were neutral, and only 15.7% perceived it to be not strict. The respondents were found to be concerned about imported IMF due to reasons ranging from delays, shortages, and virus risks. Of the 804 participants, 74.1% acknowledged these concerns, 12.9% were not sure, and the remaining 13.0% were not concerned. 

The distribution of Likert scale scores and mean values of respondents’ trust in IMF chain actors are shown in Figure 2. The results indicate medium–high trust levels in all actors, with all mean values above the midpoint of the scale. There are differences in the overall trust between the seven actors (*p* < 0.001). Respondents trust regulatory and supervisory authorities most (5.51 refers to a level between somewhat trustful and trustful), followed by traceability systems with 5.49 and organic certifying bodies with 5.42. The average trust in dairy farmers (5.11) and IMF manufacturers (5.04) was slightly above “somewhat trustful”, whereas the mean value of trust in distributors was the lowest, with minor differences between online (4.82) or offline (4.42) distributors. 

### 4.2. Consumer Preferences

Before proceeding to the estimation stage, the data collected from the online survey were organized in a long form structure as required by Stata 14.2 (StataCorp), meaning that the dataset has one row per alternative for each choice scenario that the respondents face. Thus, with 804 decision-makers choosing amongst three alternatives across eight scenarios, the dataset has 19,296 rows (804 × 3 × 8). The opt-out option was defined as a dummy variable ASC (Alternative-Specific Constant) which is equal to one in the row corresponding to the relevant alternative and zero otherwise. The dependent variable was coded as one for the chosen alternative in each scenario and zero for the non-chosen alternatives. The independent variables include alternative-specific attributes variables of product and the case-specific social-economic variables of respondents. Results for the models can be seen in Table 3, where the signs and significance values on the estimated coefficients of product attributes were consistent in both models and the random parameter logit model improved the goodness-of-fit as shown by the lower values of Log likelihood, Prob>chi^2^(6), AIC and BIC.

The significance values and the signs on the estimated coefficients of selected IMF attributes were mostly as expected, although some unexpected results emerged that can be explained. The coefficients for “functional ingredient”, “organic” and “tracible” are all positive and statistically significant at the 1 and 5% levels in both models, indicating that consumer utility could be significantly improved. Thus, respondents are more likely to purchase when the information of functional ingredients, organic labelling and traceable codes are specified on IMF packaging. Whether or not the product is traceable was by far the most influential attribute in determining the selection of IMF as the largest coefficient value in the model result. In addition, the coefficient of the opt-out option (ASC) is positive and statistically significant as expected, indicating that consumers gain more utility from choosing one of the experimentally designed IMF profiles rather than the opt-out choice. However, unexpected negative coefficients were found for the attributes of origin and price, indicating that the consumer utility could decrease with the levels of product origin and price moving up. In other words, respondents preferred a domestic brand (domestic main ingredients and domestically produced) to a foreign brand (directly purchased from a foreign country through private or cross-border e-commerce retail import, labelled in a foreign language). Respondents tend to steer away from higher-priced IMF, as the estimated coefficient of price was not significant and positive under the significance level of 5%. 

The results in Table 3 also show that the corresponding standard deviations of all product attributes except for functional ingredients are statistically significant, suggesting that consumer preferences for these attributes are heterogeneous. The random parameter logit model B1–B3 (Table 4) added the interaction terms between the case-specific demographic and perspective variables (Table 2) and alternative-specific variables (attribute variables). Dummy variable ASC illustrates the proportion of people who opted out of any of the pairs by selecting the “I would not buy either of them” option. A proportion of 41.5% (334/804) of participants selected one of the two IMF products in all eight of the choice sets, and the remainder opted out at least once. The significance of interaction coefficients ASC*Trust_Dqual and ASC*Trust_Dreguin in the result of model B1 shows that respondents are more likely to choose an alternative product rather than opt-out if they trust the quality and safety of domestically produced IMF and the governance system for the industry. Similar but more pronounced results were observed for model B3, where consumer utility was higher for a lower-priced product when they trust the quality and safety of domestically produced IMF, as well as governance. Even if the main effect of price on consumer preferences was not significant, there was also a significant interaction between price and income, where a high income would make consumer utility much higher for lower-priced products than for low-income consumers. As could be expected given the interactions in model B2, consumer utility would be higher for domestically produced products when they trust the quality and safety of domestically produced IMF, when they are confident in and satisfied with governance systems, and when they agree that COVID-19 would greatly compromise the supply of imported IMF.

## 5. Discussion

This study comes two and a half years after the first COVID-19 case was reported in late December 2019. The Chinese government implemented strict public health measures on travel restrictions, lock downs and isolation policies, all of which are ongoing. Throughout our online survey period, several of China’s major cities including Shanghai, Shenzhen and Beijing were hit with their most severe outbreaks with hundreds or thousands of cases logged. Several cases uncovered in Beijing, Zhuhai and Shenzhen were reported to be linked to contaminated goods from overseas (cold chains and international parcels) [52]. This link has been made by several local governments and media outlets but has not been scientifically established by international organizations. Chinese health officials continue to warn the public to minimise orders from overseas during the COVID-19 period and to wear protective gear when handling inbound packages [53]. China’s delivery companies have also been ordered to disinfect international packages upon arrival and hold items for about seven days before final dispatch to recipients, while postal workers exposed to international mail are required to undergo two nucleic acid tests every seven days (once every day at high-risk areas) [54]. With those COVID-19 control protocols in place, customers who purchased directly from overseas through private “daigou” cross-border e-commerce platforms were seriously discouraged from buying foreign IMF. Conventional international supply chains have also been disrupted, resulting in fluctuations in stocks and loss of customers. As described by one respondent: “Currently I prefer products produced in China for safety concerns, because we would be isolated as a close contact once the parcel was suspected of contacting a virus”. Or another, saying “We changed to domestic brands because of logistics delays; I do not like waiting that long for product from overseas”. 

The study also reveals changes in Chinese consumer behaviour that may be longer-term in nature than the shock caused by the pandemic. In line with previous findings that Chinese parents are highly quality- and safety-focused [8], this study finds that non-price attributes play a much more important role than price in affecting respondents’ purchasing IMF. Like parents everywhere, Chinese parents want the best IMF product for their children and thus select the safest (organic and traceable) products with the best nutritional value (functional ingredients). These findings on respondents’ trust in IMF chain actors are consistent with those of Wei et al. [55] who found that Chinese consumers are generally confident in the safety of domestic IMF, which has greatly increased in recent years. At the same time, consumer preferences toward domestically sourced main ingredients and domestically manufactured IMF are more pronounced for respondents that are female or with low education levels, a finding that aligns with that of Jianakoplos and Bernasek [56]. These respondents might be more likely to have concerns about the impacts of COVID-19, such as risks from imported products or product packaging. This demographic tends to pay more attention to dairy safety and is more sensitive about the related negative food scandals, which impairs their trust.

Previous studies show consumer behaviour in China as being “going for the most expensive”, which is a price–quality heuristic where consumers use price as a proxy for quality, leading to a positive correlation between prices and consumer preference [57]. This has been found in a range of contexts. Parents of a newborn babies, particularly if they are having their first and possibly only child, are very likely rely on this heuristic in deciding which brand to buy, inferring that a higher price means higher quality [46]. Parents lacking confidence in food safety are willing to pay a high price premium on the assumption that the IMF consumed by their children is reliable [7,8,12,42]. From a consumer psychology perspective, Chang [58] argued that parents tend to be price-insensitive in buying IMF because the decision involves people to whom they are very close to, and therefore depends more on their subjective feelings towards children than an objective assessment towards product attributes. Purchasing decisions are also influenced by consumers’ income and purchasing power, which has increased markedly in upper and middle strata of Chinese urban households over at least the last decade [8,59]. 

Findings in this study suggest that the price–quality heuristic is breaking down. The correlation found in this study between price and consumer preference is negative and insignificant, which implies that consumers did not evaluate price levels closely and that there was an underlying preference away from expensive products. Contrary to the findings of previous research, the insignificant price attribute suggests that price may not be a prominent factor affecting Chinese parents’ purchase decisions. A contributing factor may be income constraints in recent years. Zhang et al. [60] estimated that the COVID-19 pandemic would reduce household per capita income by 8.75% for rural residents and 6.13% for urban residents. The relaxation of the birth policy also decreases the expenditure available for each child in the family. Pei [61] showed that the two-child families are more willing to buy domestic-brand infant formula at a lower price level when compared with a one-child family due to the income constraints. 

Various past studies also show that Chinese parents favor imported IMF over domestic IMF for a number of reasons [4,7]. Consumers disappointed by scandals surrounding the safety of domestic IMF turned to imported IMF brands, especially those with a strong brand reputation. At the same time, China implemented a provisional import tax rate (5%) which is lower than the most-favoured-nation tax rate on IMF products (15%) to facilitate the imports of IMF products. It also signed free trade agreements that reduced tariffs on IMF imports from New Zealand and Australia to zero. These factors resulted in a sky-rocketing increase for IMF imports in the post-melamine period from 2008. Import volumes of IMF increased from 42,218 tons in 2008 to 302,509 tons in 2017 with an annual increase of 61.6% over the decade. This reduced the market share of domestically produced IMF from 60% in 2008 to 40.7% in 2017 [9]. 

Since that era, the Chinese government has become very concerned about the rapid decline in the self-sufficiency rate of dairy products. Government and industry responded through efforts to revitalise the domestic dairy industry and improve the quality and safety of domestic IMF to meet the nutritional needs of children (Figure 3). Efforts during in first decade (2008–2017) were directed at improving the quality and safety of IMF products through tightening regulations, systems, governance and accountability. Policies and regulations since 2018 have sought the promotion of competitiveness between domestic IMF and imported IMF. The apparent success in doing so appears to have contributed to a slowdown in year-on-year growth of imported IMF to under 10% in 2018 and 2019. It then turned negative: −3% in 2020 and −22% in 2021. In the first four months of 2022, the import volume fell further by another 6% year on year and showed no signs of recovery in May and June [62].

This study might indicate that the era where Chinese consumers preferred foreign over domestic IMF is cooling off. Consumers appear to have built confidence in the quality and safety of domestic IMF. On the premise that basic quality and safety can be guaranteed, consumers may then choose to reduce their monetary outlays (e.g., break the price–quality heuristic). Some consumers could be expected to revert back to their preferences for imported IMF products when the COVID-19 pandemic ends, but this may not necessarily be widespread. As stated by respondents: “if my daughter doesn’t react negatively to it [domestic IMF] we won’t necessarily go back to imported IMF, especially given the variety and price of domestic brands and given no end in sight to the disruption caused by the pandemic”; and another “I think the product of big domestic brand like Feihe is sufficiently safe and good quality, which should be more suitable for our Chinese babies than overseas IMF.” Thus, the pandemic shock may have forced new brand loyalties. Even if consumers don’t feel particularly loyal to a brand, they may still stick to it because of status-quo bias [63]. This is especially the case for IMF as babies can take days to adapt to a new formula [64] which parents may want to avoid. Thus, we argue that many Chinese consumers are likely to stick with the domestic IMF even when the disruption of COVID-19 subsides.

## 6. Conclusions

This study is the first choice experiment conducted since the outbreak of COVID-19 on the preferences of Chinese parents toward IMF. It found that Chinese parents continue to pay close attention to quality and safety attributes, and also found an increased confidence in domestic IMF. This is driven by supply-side developments in the Chinese dairy sector, but the process has been greatly accelerated by COVID-19. Chinese parents continue to prioritise the attributes of functional ingredients, organic labelling and traceability information. This implies a demand for further development and delivery of these attributes for both Chinese and foreign IMF companies. Considering China’s strict anti-COVID measures and logistics management, it is recommended that international IMF suppliers take measures to mitigate against further costumer loss. Investments in joint-venture farms in China are also a means by which foreign companies can capitalise on the increasing demand of domestically manufactured IMF. For all products, smart logistics and contactless delivery are recommended, especially for private daigou and cross-border e-commerce platforms. 

The study is subject to several limitations. The analysis of this paper was based on data obtained from an online survey, with self-selection for respondents with internet access that can fill in surveys online. This may bias results. For instance, we did not restrict or exclude rural consumers to participate the survey, but most of our respondents were from municipalities, provincial capitals or cities, with relatively high overall education and income levels. In addition, the choice experiment study tested attributes at only two levels (“with and without”) for functional ingredients, organic and traceability. More disaggregated levels on attributes such as nutrition contents and degree/type of traceability system were not included in this study. Thus, future research could investigate consumer preferences to IMF in China through a larger and more diverse sample that includes rural areas and tests attributes through more disaggregated indicators. A study in the medium-term future that captures longitudinal change in consumer preferences towards IMF—and the effects of the changing COVID-19 situation, the development of the domestic dairy industry and changing consumer sentiment to foreign products—would be fascinating.

## Figures and Tables

**Figure 1 foods-11-02689-f001:**
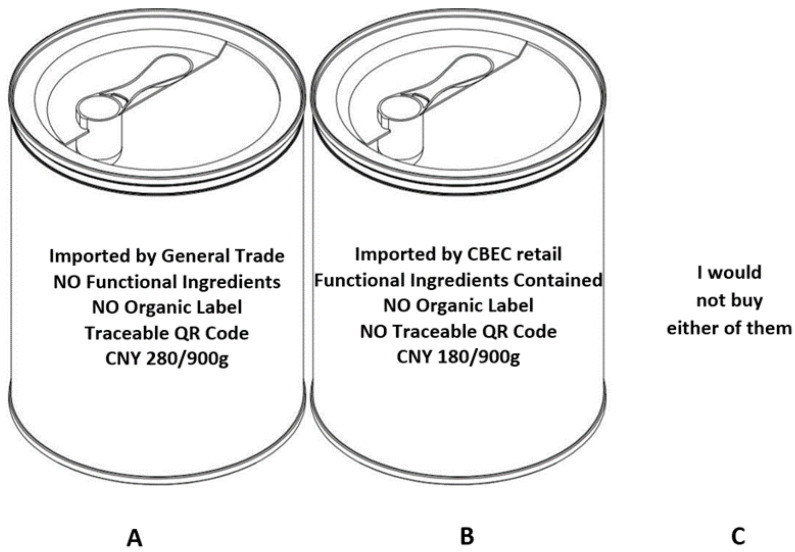
Example of one choice set ((**A**–**C**) stand for different options).

**Figure 2 foods-11-02689-f002:**
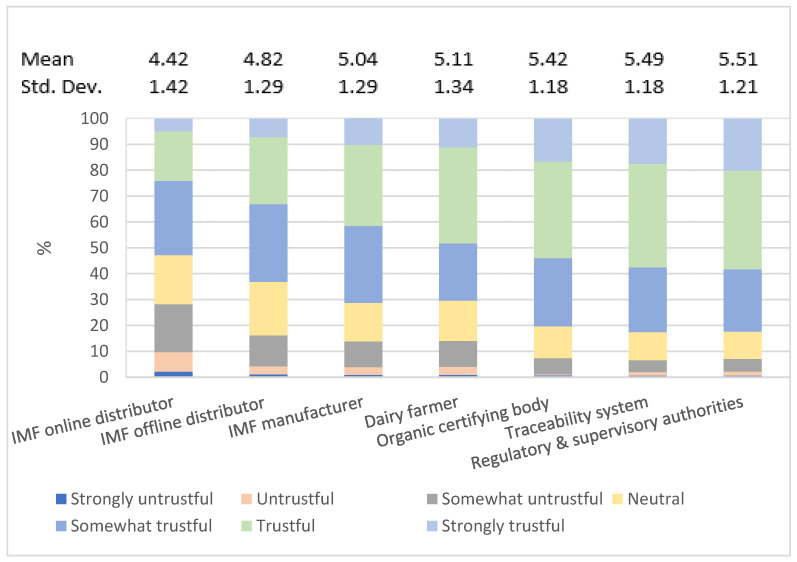
Respondents’ trust in IMF chain.

**Figure 3 foods-11-02689-f003:**
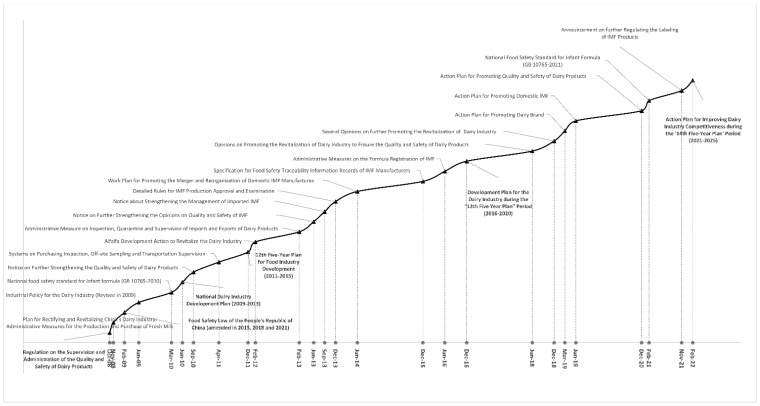
Policy and measures toward the Chinese Infant Milk Formula sector, 2008–2022. Source: Authors collected and translated from multiple Chinese official websites including State Council, Ministry of Agriculture, China Food and Drug Administration, State Administration for Market Regulation, Development and Reform Committee and Ministry of Industry and Information Technology.

**Table 1 foods-11-02689-t001:** Product attributes and attribute levels.

Product Attribute	Description	Attribute Levels
Product origin [32,34]	Origin of main ingredients (‘nai yuan’), origin of manufacturing (processing location) and country-of-purchase	1 = domestic main ingredients, produced domestically 2 = imported main ingredients, produced domestically 3 = produced overseas, imported in its original packaging, labelled in Chinese 4 = purchased from overseas through private or cross-border e-commerce retail import, labelled in a foreign language
Organic [39,40,41]	Logo or other trademark to show organic certification, regardless of Chinese or non-Chinese origin	1 = With organic label 0 = No organic label
Functional ingredients [30,33]	Functional ingredients, such as DHA/ARA for brain development, prebiotics for digestive health, lutein for vision and cognitive function etc.	1 = functional ingredients contained 0 = No functional ingredients
Traceability [31,42,43,44,45]	QR (Quick Response) code to trace any supply chain information covering milk producing, IMF processing and marketing.	1 = With traceable QR code 0 = No traceable QR code
Price [8,30,32,46]	Average price of IMF available in markets at the time of study, with intervals of CNY 50	1 = 180 CNY/900 g (around 26 USD/900 g) 2 = 230 CNY/900 g (around 33 USD/900 g) 3 = 280 CNY/900 g (around 41 USD/900 g) 4 = 330 CNY/900 g (around 48 USD/900 g)

**Table 2 foods-11-02689-t002:** Demographic frequency distribution of samples (*N* = 804).

Demographic/Perception	Category	%
Gender	Male	41.5
	Female	58.5
Age (years)	20–30	37.7
	30–40	56.7
	40–50	4.9
	Over 50	0.7
Education	Junior high	1.6
	Senior high	10.8
	College (2–3 years)	24.8
	Undergraduate	53.8
	Postgraduate & above	9.0
Family size (peoples)	3	39.6
	4	31.7
	5	17.9
	6 and more	10.8
Income (CNY)	≤100,000	11.8
	100,000–200,000	46.4
	200,000–300,000	27.4
	300,000–400,000	8.8
	400,000–500,000	2.2
	>500,000	3.4
The quality and safety of domestic produced IMF is reliable to me	Strongly disagree	3.7
	Disagree	12.6
	Somewhat disagree	15.4
	Neutral	27.7
	Somewhat agree	18.8
	Agree	11.0
	Strongly agree	10.8
China’s quality and safety supervision of IMF is the strictest in history	Strongly disagree	2.6
	Disagree	4.6
	Somewhat disagree	8.5
	Neutral	23.8
	Somewhat agree	17.3
	Agree	24.0
	Strongly agree	19.3
I would be concerned about imported IMF due to the possible delays, shortages or virus risks	Strongly disagree	1.6
	Disagree	5.1
	Somewhat disagree	6.2
	Neutral	12.9
	Somewhat agree	24.1
	Agree	26.0
	Strongly agree	24.0

**Table 3 foods-11-02689-t003:** Results of conditional logit and random parameter logit main effects.

	Conditional Logit Model (A)		Random Parameter Logit Model (B)			
Choice	Coef.	Std. Err.	Mean Coef.	Std. Err.	SD Coef.	Std. Err.
Origin	−0.3641 ***	0.0185	−0.5412 ***	0.0334	0.6257 ***	0.0358
Functional ingredient	0.1130 **	0.0450	0.1324 **	0.0553	0.1427 *	0.1641
Organic	0.4175 ***	0.0491	0.5849 ***	0.0645	0.5581 ***	0.1161
Tracible	1.2909 ***	0.0382	1.8237 ***	0.0753	1.2046 ***	0.0818
Price	−0.0048	0.0190	−0.0052	0.0248	0.2908 ***	0.0366
ASC	0.6732 ***	0.0777	1.5624 ***	0.1298	−1.8538 ***	0.1324
Log likelihood	−5851.6058		−5298.18			
Number of obs.	19,296		19,296			
LR chi2(6)	2429.34		1106.86			
Prob>chi2	0.0000		0.0000			
Pseudo R2	0.1719		-			
AIC	11,715.21		10,620.35			
BIC	11,762.42		10,714.76			

*, **, *** denotes significance level at the 10%, 5% and 1% respectively; ASC stands to Alternative-specific Constant.

**Table 4 foods-11-02689-t004:** Results of a random parameter logit model interacting the case-specific with alternative-specific variables.

	Interacting Case-Specific Variables with ASC (B1)		Interacting Case-Specific Variables with Origin (B2)		Interacting Case-Specific Variables with Price (B3)	
Choice ^1^	Coef.	Std. Err.	Coef.	Std. Err.	Coef.	Std. Err.
Origin	−0.5817 ***	0.0372	−0.8749 ***	0.1690	−0.5837 ***	0.0373
Functional ingredient	0.1247 **	0.0558	0.1379 **	0.0561	0.1266 **	0.0558
Organic	0.5980 ***	0.0659	0.6239 ***	0.0642	0.6004 ***	0.0660
Tracible	1.8464 ***	0.0759	1.8586 ***	0.0779	1.8474 ***	0.0759
Price	−0.0063	0.0252	−0.0062	0.0249	−0.1294	0.0944
ASC	−0.2639	0.6921	1.6459 ***	0.1391	−0.6796	0.4999
Gender *	−0.3000	0.2052	−0.1840 ***	0.0517	0.0214	0.0439
Education *	−0.1861	0.1497	0.1134 ***	0.0390	−0.0232	0.0323
Income *	0.0857	0.1208	0.0460	0.0304	0.0686 ***	0.0258
Trust_Dqual *	0.2364 ***	0.0658	−0.1598 ***	0.0175	0.2343 ***	0.0660
Trust_Dregu *	0.2096 ***	0.0715	−0.1127 ***	0.0184	0.2094 ***	0.0725
Impact_CV *	0.0297	0.0728	−0.0929 ***	0.0172	0.0288	0.0702
Log likelihood	−5278.5997		−5165.5757		−5277.2314	
Number of obs	19,296		19,296		19,296	
LR chi2(6)	1104.94		1060.80		1118.49	
Prob>chi2	0.0000		0.0000		0.0000	
AIC	10,593.20		10,367.15		10,590.46	
BIC	10,734.82		10,508.77		10,732.08	

^1^ Other variables omitted for lack of statistical significance were age and family size; *, **, *** denotes significance level at the 10, 5 and 1% level respectively; ASC stands to Alternative-specific Constant; Trust_Dqual stands for Trust of Domestic product quality; Trust_Dregu stands for Trust of Domestic industry regulation; Impact_CV stands for Impacts of COVID-19.

## Data Availability

Data is contained within the article.

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
