# Peer review of "Evidence from a Choice Experiment in Consumer Preference towards Infant Milk Formula (IMF) in the Context of Dairy Revitalization and COVID-19 Pandemic"

_foods, 2022, doi:10.3390/foods11172689_

Round 1
Reviewer 1 Report
This is an interesting paper capturing changes in consumption behaviour that emerged during the COVID-19 pandemic. The study lacks a certain degree of detail that is needed for the reader to follow the paper. Below, I list a few recommendations on how to improve the paper.
Title:
Grammar is odd. I suggest to keep the first part and drop what has been added after the :
Or revise as follows: Evidence from a choice experiment in consumer preference to-2 wards infant milk formula (IMF) in the context of dairy revitalization and COVID-19 Pandemic
L 103: consumerS
“However anecdotal evidence and various studies [33] suggest that consumer preferences, perceptions and habits may have changed in the era of the COVID-19.”
This is a bold statement that requires more than one source (“various studies”). Please explain what the specific changes were.
The aim is phrased as follows: “This paper aims to provide up-to-date empirical evidence on change in consumer perceptions and confidence to IMF in the COVID-19 era.”
In my view, when the focus is set on “change”, more information needs to be provided on the current situation. Alternatively, the aim can be rephrased as “evidence on current consumer perceptions”, making it clear that the study investigates the situation as is now, during the pandemic.
Please explain how you receive 42×23=128 virtual 126 product profiles
The eight choice sets should be explained / defined.
The sampling is confusing. In the beginning of the paper, you state that you aimed for parents with children under 3. In the discussion, however, the skewed age distribution in discussed together with “grandparents taking care of grandchildren”, which, however, does not fit the sample description of “parents”. Please revise.
This seems relevant for a data paper, but not a research paper:
Before proceeding to the estimation stage, the data collected from the online survey were organized in long form structure as required by Stata 14.2 (StataCorp), meaning that the dataset has one row per alternative for each choice scenario that the respondents face.
Revise sentence (grammar seems odd):
Lancaster's characteristics demand theory and McFadden’s random utility theory allow this study divided respondent's utility into an observable deterministic component and an unobserved random component as shown in equation (1) below.
Incentive fee: please provide amount in $ for readers not familiar with the Chinese currency
Figure 2: It would be helpful if actors were sorted in regard to mean values. This way, the reader can easily spot the most important and least important actors.
When you provide percentages (y-axis), make sure it ends with 100%, otherwise it is confusing.
Table 3: what does “title 1” mean?
Table 4: ACS and ASC and other variables that are not self-explanatory need to be explained in the table notes.
Discussion:
In the discussion section, qualitative responses from respondents are introduced. To me, it remains unclear how these answers were obtained. Was this part of the survey? If so, this should be described more clearly in the methods section.
The discussion section is lacking a discussion of the study’s limitations. Further, it also lacks a discussion of the present work in the context of other research.
Figure 3 is a really nice visualisation. Unfortunately, the font is too small to read. Please revise. Further, it might be easier to read and interpret if the years were somehow made visible (e.g. vertical lines)
Conclusion:
Discussion of limits should be moved to discussion section.
I see the point that grandparents can be involved in taking care of their grandchildren. However, I would assume that parents make decisions on what formula to use. In similar line, the age skewed sample should be unproblematic, as parents tend to be younger and this would be the group you are interested in as they make the decisions.
One limitation I see is that there was no restriction for participation in the survey. In my view, the survey should have been addressed to parents.
Author Response
We are grateful for your comments which have helped us improve the quality of the paper. The responses are highlighted in blue text.
Please see the attachment.

Reviewer 2 Report
The authors of the study are to be congratulated on the choice of the research topic and the idea of carrying out the scientific study (planning and carrying it out). The issues addressed in the article are very interesting and of interest to the reviewer.
Infant feeding is not only a time of eating, but also a moment of the first social interactions and the strengthening of the bond between mother and child. During this time, the toddler satisfies many other needs that determine its proper development and well-being. The proximity of the mother, the sound of her voice, the eye contact - all this, combined with the suckling activity, is a source of physical pleasure and a sense of security for the little one.
In my opinion:
- The manuscript uses a clear introduction;
- The material and research methods were described in detail;
- The study was properly planned;
- Appropriate analytical methods were selected to analyse the results;
- The results of the analyses were presented in graphical and tabular form.
Please respond to my concerns:
1. Please list the factors, relevant from the authors' point of view, which, in addition to declining family income during the COVID-19 pandemic, cause the choice of cheaper IMF?
2. What has caused women in China to stop breastfeeding their children in recent years?
3. Are Chinese women encouraged to breastfeed rather than give IMF milk early?
4. Are there social campaigns targeting mothers of young children in China to popularise domestic IMF?
Author Response

(The authors gave the same response as above.)

Reviewer 3 Report
Some specific comments to your work:
It would have been interesting to have a greater number of people surveyed and also to really know what IMF they are giving their children since it is not shown in the study.
Introduction
The objectives would be only one (The aim is document Chinese parents’ preferences toward IMF since the outbreak COVID-19) and not two as the authors indicate, and it would also be unnecessary to indicate in the introduction to name the sections of the work.
Materials and methods
It would be important to know the questionnaire used.
Results
Lines 233-235 and 295-298 These sentences should go to the discussion section and not to the results section:
“These findings are consistent with those of Wei et al. [47] who find that Chinese consumers are generally confident in the safety of domestic IMF, which has greatly increased in recent years”.
“At the same time, consumer preferences toward domestically sourced main ingredients and domestically manufactured IMF are more pronounced for respondents that are female or with low education levels, a finding hat aligns with that of Jianakoplos and Bernasek [48]”.
Discussion
Lines 314-316: This sentence is based on journalistic information and not with scientific rigor.
Lines 392-393 With a population sample of 804, this sentence is too daring: “This study demonstrates that the era where Chinese consumers preferred foreign 392 over domestic IMF has come to an end”.
Author Response

(The authors gave the same response as above.)

Round 2
Reviewer 3 Report
Thanks to the authors for considerings my comments.